# Towards a RINA-Based Architecture for Performance Management of Large-Scale Distributed Systems †

**Peter Thompson \*** and **Neil Davies**

Predictable Network Solutions Limited, Stonehouse, Gloucestershire GL10 2PG, UK; neil.davies@pnsol.com
* Correspondence: peter.thompson@pnsol.com
† This paper is an extended version of our paper published in The 23rd Conference on Innovation in Clouds, Internet and Networks (ICIN 2020), Paris, France, 24–27 February 2020.

**Abstract:** Modern society is increasingly dependent on reliable performance of distributed systems. In this paper, we provide a precise definition of performance using the concept of quality attenuation; discuss its properties, measurement and decomposition; identify sources of such attenuation; outline methods of managing performance hazards automatically using the capabilities of the Recursive InterNetworking Architecture (RINA); demonstrate procedures for aggregating both application demands and network performance to achieve scalability; discuss dealing with bursty and time-critical traffic; propose metrics to assess the effectiveness of a performance management system; and outline an architecture for performance management.

**Keywords:** RINA; performance; quality attenuation; distributed system; system scalability; ΔQ

---

## 1. Introduction

Online services, implemented by distributed computing systems, have displaced traditional ones in many areas: shopping; entertainment; social interaction; banking; and public services, to name but a few. Take-up of such online services is widespread, making maintenance of alternative service provision using manual processes increasingly uneconomic. Consequently, modern societies are becoming ever more dependent on the reliability of these online services (a trend that has massively accelerated during the Covid-19 pandemic), and it is anticipated that access to such services will have a transformative effect in developing economies (which would therefore also become dependent on them).

At the same time, there are growing societal concerns: firstly that such online services are not sufficiently reliable [1]; and secondly that implementing this trend using a traditional client–server model places too much power in the hands of a small number of service providers. This is fuelling interest in service architectures that are less dependent on centralised control and management, for example distributed ledgers such as blockchains [2].

Combining high reliability with independence from centralised management and control requires a considerable degree of autonomous self-management on the part of the distributed system. This is enabled by developments such as software-defined networking (SDN) [3], which increase the dynamic configurability of networks. However, such self-management and re-configuration must take account of the emergent performance (which will otherwise inevitably degrade) and the resource cost (which will otherwise inevitably grow).

In this paper, we outline an approach to distributed system performance management based on a robust measure that is applicable at all layers of the system, from application down to packet transfer. We relate the requirements of such performance management to the capabilities of RINA (Recursive InterNetworking Architecture) [4], a re-conceptualisation of 'networking' as fundamentally

about communication between processes rather than transmission of data. RINA unifies protocol 'layers' into a repeated structure with common mechanisms specialised by 'policies', thereby achieving vast simplification, in particular for network management. Conversely, RINA distinguishes key features that have been catastrophically conflated in the current TCP/IP architecture, such as naming vs. addressing, and control vs. data-transfer [5]. RINA also provides nested scopes called DIFs within which management of performance is feasible, and control loops can be short enough to address congestion in a timely fashion [6]. It therefore provides a solid foundation for a performance management architecture, as discussed in Section 5.

For a performance management system to be sustainable, it also needs means to monitor its effectiveness, and an 'economics' to incentivise appropriate usage, for example through charging to reflect the differential costs of delivering system performance (out of scope here).

*Structure of the Paper*

Following this introduction, the paper is structured as follows:

- In Section 2, we refine the notion of 'performance' using a precise measure called quality attenuation (referred to in other literature as 'quality impairment' or 'quality degradation');
- In Section 3, we apply this to managing the performance of distributed systems;
- In Section 4, we consider overbooking and correlation hazards to the delivery of good performance;
- In Section 5, we discuss how the features of RINA assist in managing performance hazards;
- In Section 6, we propose metrics to assess the effectiveness of a performance management system;
- In Section 7, we briefly outline an architecture for performance management;
- In Section 8, we draw some conclusions and consider directions for further work.

This paper is based on [7], with the addition of:

1. Background information on RINA and detail on how it interacts with performance management;
2. More detail on the decomposition of $\Delta Q$;
3. A description of the process of aggregating QTAs;
4. A new section on dealing with bursty traffic;
5. Expanded conclusions and discussion of directions for future work;
6. An appendix showing how $\Delta Q$ can be calculated from application behaviour.

## 2. Defining Performance

Managing performance in a distributed system requires a relevant definition of 'performance' and a means of relating this to controllable parameters of the system. This can be tackled in a variety of ways for small and/or specialised systems [8]. However, managing performance at a large scale across a network serving many users with highly varied and constantly changing applications introduces additional constraints. In particular, there is a tension between the complexity of the performance management and its effectiveness for individual users and applications.

To avoid an explosion of complexity (so that the cost of running the system is contained), the performance management must be firstly, generic, i.e., applicable to all types of applications, and, secondly, operable on coarse-grained aggregates. To be useful on an individual user/application level, however, it must also be possible to dis-aggregate performance management on a fine-grained basis. Finally, any control loops involved, e.g., for management of congestion, must operate over appropriately bounded timescales, as determined by the constraints of basic control theory [9].

### 2.1. Service Performance

We define a *service* as a distributed process generating outcomes, where an outcome is something that can be observed to begin and end (since the service is distributed, the beginning and end of

an outcome typically occur at different locations). Outcomes may be unique/intermittent (e.g. web page download), or repeated on a regular basis (e.g., a video client/server presenting a sequence of frames on a display). Such outcomes must typically meet some timeliness constraint in order to be useful, and occur at a certain rate or with a certain total volume. Where the service depends on shared resources (which is usually necessary to manage costs) the performance will be statistical, for example X% of outcomes complete within Y seconds, with Z% failing to complete at all; in the video example, this defines the occurrence of playback glitches or pauses. The combination of rate/volume and success/timeliness is the *performance* of the service.

Performance metrics for other services include:

- VoIP: audio glitches per call minute;
- Web page download: time from request to rendering of the initial part of the web page;
- Web page rendering: time from requesting the page to rendering of the last element;
- Online gaming: enjoyment-impacting game response delays per playing hour;
- Bank transfers: time from requesting transfer to funds appearing in the other account.

It is worth noting that outcomes are affected by a variety of factors. For example, a bank transfer's time to complete may just be a question of network/application performance (e.g., an online transaction of a sufficiently small amount one standard account and another inside the same bank group) or it may depend on issues outside the distributed application's remit (e.g., if it has to go through money laundering checks or is coming from a delayed access account). Such external factors are beyond the scope of performance management as defined here.

This approach to performance also applies to 'real world' services such as parcel delivery, but our focus here is on electronic services mediated by packet networks.

### 2.2. Quality Attenuation: ΔQ

'Performance' is typically considered as a positive attribute of a service. However, a perfect service would be one without error, failure or delay, whereas real services always fall short of this ideal; we can say that their quality is *attenuated* relative to the ideal. Quality attenuation encapsulates all types of failure or delay in delivering an outcome, and is denoted here by the symbol ΔQ. We recast the problem of managing performance as one of maintaining suitable bounds on ΔQ [10] while delivering the required rate or volume of service outcomes.

This is an important conceptual shift because, whereas 'performance' may seem like something that can be increased arbitrarily, ΔQ (rather like noise) is something that may be minimised but never eliminated completely. Indeed, some aspects of ΔQ, such as the time required for signals to propagate between components of a distributed system, cannot be reduced below a certain point. This transforms performance management into a tractable engineering problem.

Note that the issue of capacity is now a secondary consideration: it simply represents the level of applied load beyond which quality attenuation increases rapidly. Maintaining a bound on ΔQ therefore implies respecting capacity constraints. Respecting such constraints is necessary but not sufficient to maintain bounds on ΔQ.

Typical audio impairments that can affect a telephone call (such as noise, distortion and echo) are familiar; for the telephone call to be fit for purpose, all of these must be sufficiently small. Analogously, the attenuation of the translocation of information (information does not 'move', but simply becomes available at a different location, generally through a process of repeated copying) between components of a distributed application must be sufficiently bounded for it to deliver fit-for-purpose outcomes; moreover, the layering of network protocols isolates the application from any other aspect of the packet transport. This is such an important point that it is worth repeating: the great achievement of network and protocol design (up to and including RINA) has been to hide completely all the complexities of transmission over different media, routing decisions, fragmentation and so forth, and leave the

application with only one thing to worry about with respect to the network: the quality attenuation that its packet streams experience, ΔQ.

ΔQ is 'conserved' in the sense that any delay in delivering an outcome cannot be undone, nor can exceptions or failures be reversed (at least not without incurring more delay). Thus, while different aspects of ΔQ can be traded to some degree, ΔQ as a whole cannot be reduced without changing the loading factor [11].

### 2.2.1. Mathematical Representation of ΔQ

In capturing the deviation from ideal behaviour, ΔQ incorporates both delay (a continuous variable) and exceptions/failures (discrete variables). This can be conveniently modelled using improper random variables (IRVs), i.e., random variables whose total probability is $\leq 1$ (these are introduced on p146 of [12] as 'defective' random variables.). If we write $\Delta Q(x)$ for the probability that an outcome occurs in a time $t \leq x$, then we define the 'tangible mass' (also known as the zeroth moment of the probability distribution) as:

$$T(\Delta Q) = \lim_{x \to \infty} \Delta Q(x) \tag{1}$$

The 'intangible mass', which is simply $1 - T$, encodes the probability of exception or failure (which might simply be the dropping of a packet due to transient buffer overflow).

We can define a partial order [13] on such variables, in which the 'smaller' attenuation is the one that delivers a higher probability of completing the outcome in any given time:

$$\forall x \cdot \Delta Q_1(x) \leq \Delta Q_2(x) \iff \Delta Q_1 \geq \Delta Q_2 \tag{2}$$

Note that this partial order has both a 'top' element (all outcomes occur instantaneously) and a 'bottom' element (all outcomes fail).

ΔQ represents the *instantaneous* distribution of delay and loss, which can be thought of as the Bayesian expectation of what would happen to an outcome started at any particular moment, and so is a function of the time the outcome starts. In many cases, ΔQ does not vary significantly over time, in which case we call it 'stationary'. When it does vary in time (i.e., is non-stationary), we are typically interested in the largest ΔQ over some period of interest, which we write $\lceil \Delta Q \rceil$.

Note that, where outcomes are parameterised in some way (e.g., size of an SDU to be transferred), ΔQ will also be a function of such parameters.

### 2.2.2. Components of Quality Attenuation

Quality attenuation arises from a variety of causes, which can be broadly categorised as:

- Physical constraints: for example, an outcome involving transmission of information from one place to another cannot complete in less time than it takes light to travel the distance between them;
- Technological constraints: for example, a computation requiring a certain number of steps depends on a processor executing the corresponding instructions at a certain rate; and communicating some quantity of data takes time depending on the bit-rate of the interfaces used;
- Resource sharing constraints: when an outcome depends on the availability of shared resources, the time to complete it depends on the other use of those resources (and the scheduling policy allocating them).

The first two sets of constraints tend to be determined by system design and implementation choices; the third varies depending on overall demand.

This leads to a natural decomposition of ΔQ into basic components:

$\Delta\mathbf{Q}_{|G}$ The infimum of outcome completion times with the effects of outcome size and resource sharing removed; in the case of packet transmission, it can be thought as the minimum time taken for a hypothetical zero-length packet to travel the path;

$\Delta\mathbf{Q}_{|S}$ A function from the outcome size to the additional time to complete it, with the infimum time and the effects of resource sharing removed;

$\Delta\mathbf{Q}_{|V}$ The additional time to complete the outcome due to sharing of resources, with the infimum time and the effects of outcome size removed.

In general, each of these is an IRV, but in some circumstances a simpler representation is possible. For example, in packet networks with stable routing, the minimum time for a packet to transit a particular network path is essentially constant and so $\Delta Q_{|G}$ can be reduced to a number, with dimensions of time. Likewise, where transmission interfaces have fixed bit-rates and are work-conserving, the relation between the size of an SDU and the time to transmit it is essentially linear, and so $\Delta Q_{|S}$ can also be reduced to a number with dimensions of time/bit. $\Delta Q_{|V}$ is intrinsically a distribution, but can be partially characterised using moments or centiles.

The combination of $\Delta Q_{|G}$ and $\Delta Q_{|S}$, written $\Delta Q_{|G,S}$, is sometimes referred to as 'structural $\Delta Q$', since, in networks, it is a consequence of the topology and technology choices in the construction of the network.

Outcome failure such as packet loss is an intrinsic part of $\Delta Q$. This follows the same decomposition:

- A failure rate that is intrinsic, independent of both outcome size and load, is the intangible mass of $\Delta Q_{|G}$;
- A failure rate that depends on the outcome size is the intangible mass of $\Delta Q_{|S}$;
- A failure rate that depends on the load is the intangible mass of $\Delta Q_{|V}$.

### 2.2.3. Compositionality of Quality Attenuation

Services are typically implemented in layers, in which an outcome at one layer is dependent on one or more outcomes at a lower level. For example, in RINA, communication of an N+1-layer SDU is implemented by transmitting one or more PDUs at an N-layer, subject to local policies for security, flow-control, etc. This dependency translates into a relationship between the $\Delta Q$ of the outcome of interest and the $\Delta Q$s of the lower-level outcomes on which it depends. This relationship may be complex and nonlinear, but will typically be monotonic (in the presence of work-conservation), in that a larger $\Delta Q$ for one of the lower layer outcomes will imply a larger $\Delta Q$ for the higher layer one also.

Consider the simple case that an N+1-layer SDU is simply wrapped into an N-layer PDU: in this case, the $\Delta Q_{|G}$ of the N+1 layer is that of the N-layer plus the time to transmit the additional header of the N-layer (the size of the N-layer header times the $\Delta Q_{|S}$ of the N-layer).

Figures 1a,b show examples of the relationship between the $\Delta Q$ of a lower-layer outcome (in this case, the delay and loss probability of individual packets) with that of two higher-layer outcomes (in this case the 95th centile of the time to complete a 20kB HTTP transfer and the PESQ score of a G.711 voice call (this diagram courtesy of CERN [14])).

Where the behaviour of the application/protocol is known, the relationship is calculable a priori. A simple worked example of this is given in Appendix A.

Furthermore, quality attenuation is 'additive' within a single layer of a system. When an outcome depends on a sequence of steps (such as computations or forwarding of PDUs), the $\Delta Q$ of the whole outcome is the 'sum' of the $\Delta Q$s of the individual steps (if the $\Delta Q$s are independent IRVs the summation operation is simply convolution). This additivity distributes across the decomposition into components discussed in Section 2.2.2 above. This 'compositionality' is illustrated in Figure 2.

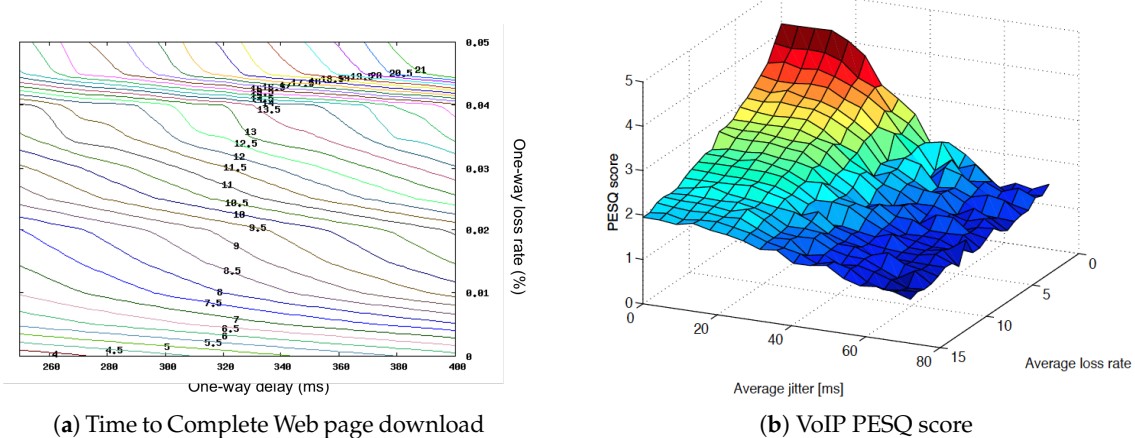

(**a**) Time to Complete Web page download　　　　　(**b**) VoIP PESQ score

**Figure 1.** Application performance contours.

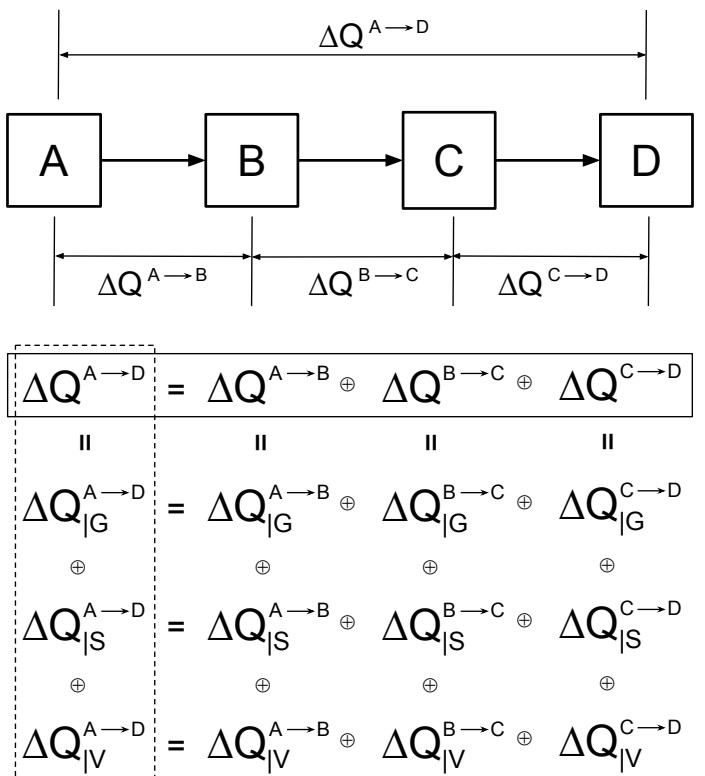

**Figure 2.** Additivity of ΔQ by components.

It is this compositionality that makes quality attenuation an effective measure for managing the performance of a large distributed system. It allows performance management to be devolved to subsystems and lower layers, by means of quality attenuation 'budgets'. Compositionality and monotonicity guarantee that keeping lower-layer and subsystem attenuations smaller than their budgets (as defined by Equation (2)) ensures that the attenuation of the overall outcome is smaller than the requirement, which is to say that its performance target is met.

RINA provides a natural framework for managing this composition, since it defines a hierarchy of scopes of control through the layering of DIFs (see Section 5).

## 3. Managing Performance

In a system with statistically shared resources, performance necessarily becomes statistical, and managing performance becomes a matter of controlling probabilities. It is not sufficient to observe performance failures and react to them because:

1.    Observing a single failure gives little information about the relevant probabilities;
2.    Waiting for a statistically significant number of failures makes the system response too slow to avoid undesirable service degradation.

A performance management system must therefore deal with probabilities that service outcomes may not be successful and/or timely. Estimating and managing such probabilities is the topic of this section.

### 3.1. Measuring Quality Attenuation

Management requires measurement, so we need a procedure for measuring quality attenuation. This requires, firstly, capturing the elapsed time between the leading and trailing edges of an outcome (e.g., a PDU starting to be sent and being completely received), and, secondly, applying appropriate statistical analysis to a sequence of such values.

The former can be implemented in two ways:

1.    Timestamping PDUs (measuring quality attenuation within the N-layer);
2.    Introduce additional SDUs containing a timestamp (measuring quality attenuation across the N-layer using a special-purpose N+1-layer DIF).

This is illustrated in Figure 3. The second approach has the advantage of providing measurements even when other flows are idle, but the disadvantage of adding load. Alternatively, the timestamp PDUs could be generated by the layer management components of the N-layer DIF (thanks to an anonymous reviewer for this observation).

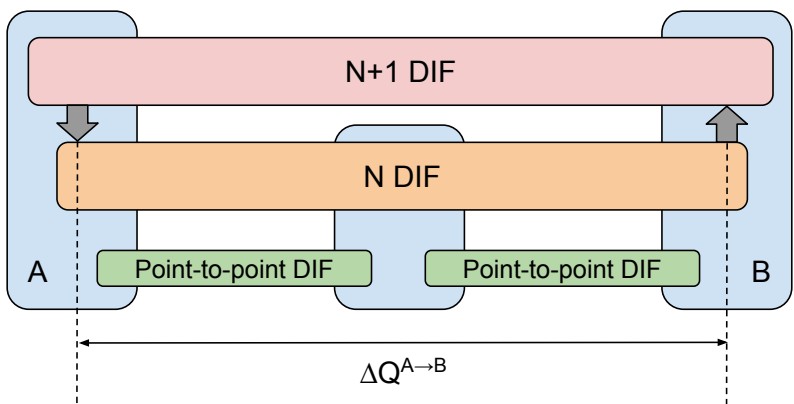

**Figure 3.** Measurement of $\Delta Q$ across DIFs.

We can extract the components of $\Delta Q$ as defined in Section 2.2.2 by statistical analysis of a set of sample measurements. By varying the size of the PDUs and fitting a line to the minimum delay per PDU size, we obtain an estimate for $\Delta Q_{|S}$ (provided resources along the path are occasionally idle so there will be some PDUs that experience no delay due to resource contention). Extrapolating this to a PDU size of zero provides an estimate for $\Delta Q_{|G}$.

When these factors are subtracted from the original delays, what remains is $\Delta Q_{|V}$. This is illustrated in Figure 4, which shows a series of PDU delivery times sorted by PDU size, and the line fitted to the minimum delays.

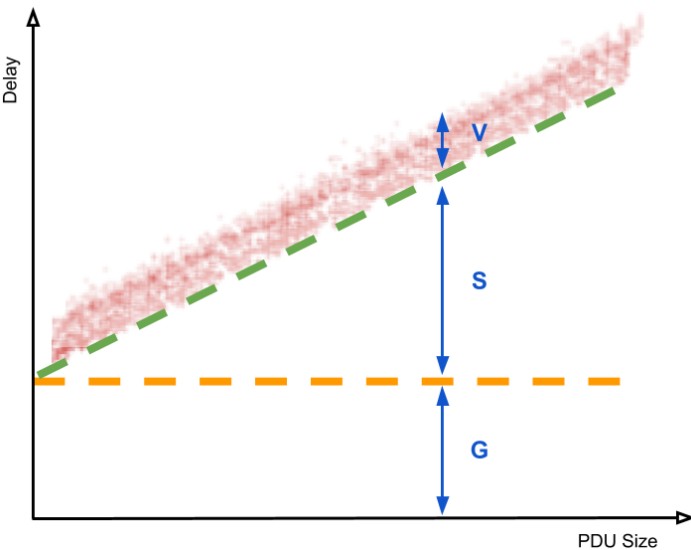

**Figure 4.** Decomposition of ΔQ.

### 3.2. Setting a Performance Bound

We can use the partial ordering relation of Equation (2) to define a notion of 'good enough' performance. Suppose we specify that 50% of service outcomes should occur within 5 s, 95% within 10 s, 99% within 15 s, with a 0.1% chance of not completing, then we could represent this as a CDF as shown in the blue line of Figure 5a. If the measured response is the black curve of Figure 5b, this satisfies the ordering relation of Equation (2), and so the delivered ΔQ is strictly less than the specified bound, i.e., it is 'good enough'.

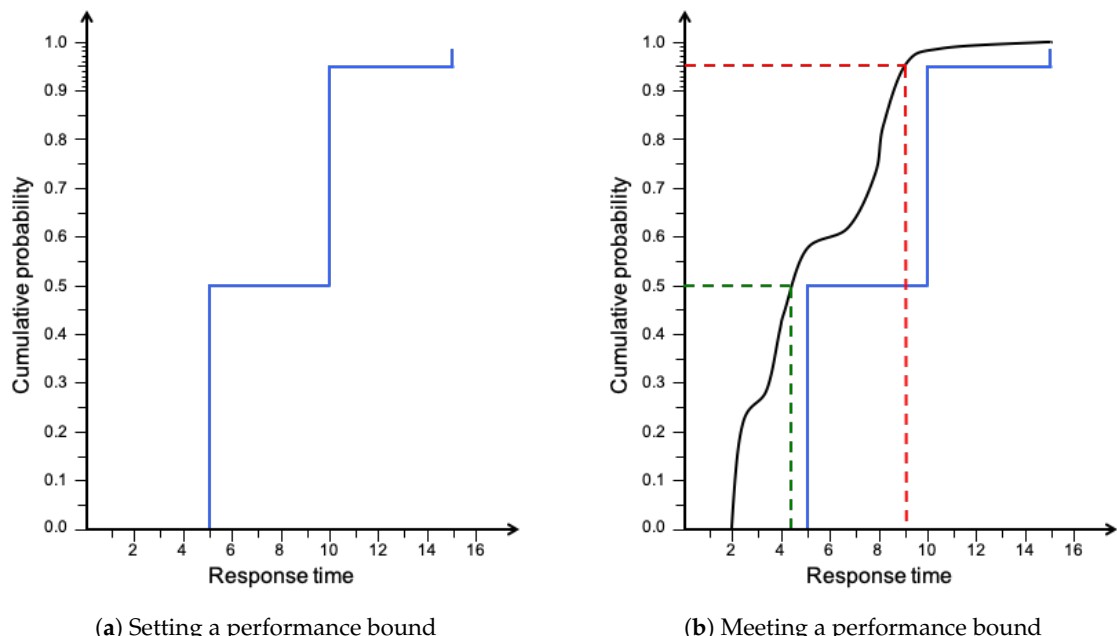

(**a**) Setting a performance bound　　　　　(**b**) Meeting a performance bound

**Figure 5.** Setting and meeting a performance bound.

### 3.3. Managing Demand: Quantitative Timeliness Agreements

Outcomes are rarely one-off events, but tend to be required repeatedly. As noted above, outcomes may have a 'size', so the combination of outcome size and rate determines the total load. Since any

given system will have only finite resources, it cannot deliver outcomes with bounded attenuation against an unbounded load. Performance management thus requires management of load as well as $\Delta Q$. This can be captured as a 'quantitative timeliness agreement' (QTA) which comprises bounds on both the applied load and the required attenuation. A QTA can thus be characterised as a requirement for a 'quantity of quality'. Decomposing QTAs through layers and/or across subsystems thus provides a complete framework for managing the performance of a distributed system. RINA provides tools for this in the concept of a 'QoS-cube' [15].

### 3.4. Aggregation of QTAs

Aggregation occurs when multiple streams are multiplexed onto a common bearer whose data rate is at least as great as the largest requirement of the individual streams.

#### 3.4.1. Allocation to Peak

One common approach to such aggregation is to construct a common bearer whose capacity is the sum of all the capacities of the individual streams. Assuming fairness of delivered $\Delta Q$ to the component flows within an aggregate, this treats aggregation akin to a simple arithmetic operation. Providing the constituent streams are not adversely correlated (e.g., not synchronised by an external clock or triggered by the same event), then this is a 'performance safe' action as the $\Delta Q$ of the individual streams in the aggregate will be no worse than the $\Delta Q$ those streams would have received if they had been carried each by their own individual bearer.

#### 3.4.2. Service Time Effects of Aggregate Bearer

An aggregate bearer, having a higher service capacity than any constituent bearer will present a lower $\Delta Q_{|S}$ to such aggregated flows than they would have experienced before aggregation (although, depending on the constituent streams' levels of mutual correlation, there may be more $\Delta Q_{|V}$). This has the potential to reduce the portion of the end-to-end $\Delta Q$ budget consumed, a budget that can be used elsewhere, and/or in satisfying other QTA requirements.

#### 3.4.3. Statistical Multiplexing Gain in a $\Delta Q$ Setting

There is also the statistical multiplexing gain which we are now in a position to quantitatively exploit, not just from the aspect of the delivered $\Delta Q$ to the streams themselves, but also from the perspective of resource allocation and scheduling/queuing configuration.

Statistical multiplexing gain is an attribute of the size of the population of sources, not their data rates. This effect is illustrated in Table 1. Here, we consider voice calls which are using an encoding generating 50 packets per second, with silence suppression; speech is present at a typical duty cycle of 38.53% (which allows for turn-taking during a conversation). This is the load specification of the QTA. The $\Delta Q$ requirement of the QTA includes a probability of discarding a packet $\leq 1 \times 10^{-3}$. Thus, the first question with regard to aggregation is what capacity is needed to fulfil this QTA for a given number of calls? We can treat this as a simple problem in combinatorics: what is the probability of *N* from *M* being active at any time?)?

It is clear that for a *single* voice call an allocation to at least the peak requirement is needed (though the issue of serialisation delay may apply, as discussed in Section 4.1.2) as, when speaking, the total data rate is required. However, as the call population increases, the allocation can rapidly approach the mean while keeping within the desired QTA.

Other types of traffic offer different opportunities for statistical multiplexing gain, depending on their traffic characteristics and correlation behaviour.

**Table 1.** Aggregation of on-off VoIP streams

| (pkts/sec) | | (pkts/sec) | | | | Risk to StdDev | |
| --- | --- | --- | --- | --- | --- | --- | --- |
| | | | | | | $1.00 \times 10^{-5}$ | 4.265 |
| on Rate | Duty Cycle | Equiv Rate | Variance | # Std Dev | | $1.00 \times 10^{-4}$ | 3.72 |
| 50 | 38.53% | 19.265 | 19.265 | 3.1 | | $1.00 \times 10^{-3}$ | 3.1 |
| | | | | | | $1.00 \times 10^{-2}$ | 2.33 |
| No of Sources | Peak Rate | Average Rate | Variance of Estimator | Interesting Centile | Allocated Rate to Achieve | Ratio: Rate/Average | Allocation |
| 1 | 50 | 19.27 | 19.27 | 13.61 | 32.87 | 1.70628 | 32.87 |
| 2 | 100 | 38.53 | 9.63 | 9.62 | 57.77 | 1.49942 | 57.77 |
| 3 | 150 | 57.80 | 6.42 | 7.86 | 81.36 | 1.40777 | 81.36 |
| 4 | 200 | 77.06 | 4.82 | 6.80 | 104.27 | 1.35314 | 104.27 |
| 5 | 250 | 96.33 | 3.85 | 6.09 | 126.75 | 1.31586 | 126.75 |
| 10 | 500 | 192.65 | 1.93 | 4.30 | 235.68 | 1.22335 | 235.68 |
| 15 | 750 | 288.98 | 1.28 | 3.51 | 341.67 | 1.18236 | 341.67 |
| 20 | 1000 | 385.30 | 0.96 | 3.04 | 446.15 | 1.15793 | 446.15 |
| 25 | 1250 | 481.63 | 0.77 | 2.72 | 549.66 | 1.14126 | 549.66 |
| 50 | 2500 | 963.25 | 0.39 | 1.92 | 1059.46 | 1.09988 | 1059.46 |
| 60 | 3000 | 1155.90 | 0.32 | 1.76 | 1261.30 | 1.09118 | 1261.30 |
| 70 | 3500 | 1348.55 | 0.28 | 1.63 | 1462.39 | 1.08442 | 1462.39 |
| 80 | 4000 | 1541.20 | 0.24 | 1.52 | 1662.90 | 1.07896 | 1662.90 |
| 90 | 4500 | 1733.85 | 0.21 | 1.43 | 1862.93 | 1.07445 | 1862.93 |
| 100 | 5000 | 1926.50 | 0.19 | 1.36 | 2062.56 | 1.07063 | 2062.56 |
| 125 | 6250 | 2408.13 | 0.15 | 1.22 | 2560.25 | 1.06317 | 2560.25 |
| 150 | 7500 | 2889.75 | 0.13 | 1.11 | 3056.39 | 1.05767 | 3056.39 |
| 175 | 8750 | 3371.38 | 0.11 | 1.03 | 3551.37 | 1.05339 | 3551.37 |
| 200 | 10,000 | 3853.00 | 0.10 | 0.96 | 4045.42 | 1.04994 | 4045.42 |
| 250 | 12,500 | 4816.25 | 0.08 | 0.86 | 5031.39 | 1.04467 | 5031.39 |
| 300 | 15,000 | 5779.50 | 0.06 | 0.79 | 6015.17 | 1.04078 | 6015.17 |

## 4. Managing Performance Hazards

A performance hazard occurs when the probability bounds on delivering service outcomes are breached. Note that this does not mean that any service outcomes actually fail; for example, the service in question might be inactive at the time. The service needs to be active for the hazard to be *armed*. Since the performance management system typically cannot know when a service may become active, its role is to keep performance hazards down to an acceptable level, and to mitigate them if and when they mature.

We consider, firstly, performance hazards due to overbooking, and then discuss further hazards due to supply variation, demand variability, and correlated load.

### 4.1. Managing Overbooking Risks

We have so far outlined an approach to delivering statistical performance bounds against well-defined and bounded loads. This has some practical limitations:

1. Loads tend to be intermittent, and thus this approach potentially wastes considerable resources;
2. Available resources may vary, for example when communicating across a wireless bearer, or when subject to a denial-of-service attack.

The typical approach to deal with the first problem is to 'overbook' resources, allowing more potential load than the system can deliver attenuation bounds for; this is similar to the situation that the second problem creates when available resources drop. Thus, it is not sufficient simply to create and decompose QTAs; the system must also track the risk of such QTAs being breached in operation [16].

### 4.1.1. Capacity, Urgency and Overbooking

Overbooking is typically considered in relation to capacity; having sufficient capacity means being able to satisfy the aggregate demand on average. However, when considering bounding quality attenuation, we must also be concerned with *schedulability*, the ability to service the instantaneous demand within the required $\Delta Q$ bounds.

We can define a 'natural urgency' for a stream, where the outcome time is the same as the fluid flow approximation service rate. For instance, a load of a 1 kB PDU once a minute ($\approx$133 bps) would have a natural urgency (the time bound that the average rate would imply) of 60 s, and the natural urgency for a 500B PDU at the same net data rate would be 30 s. Delivering PDUs with less delay than this, consuming more than the natural amount of the urgency budget, creates a 'urgency debt' that must be paid for, because, as noted in Section 2, $\Delta Q$ is conserved. This is either by:

- other traffic getting service worse than its natural urgency;
- this stream (or others) experiencing loss; or
- under-utilisation of the capacity.

This is, in some ways, a restatement of the notion of effective bandwidth [17] (where the economics and pricing structure is the key driver in the context of a concrete deployment), but framed in terms of the costs of supporting the demanded $\Delta Q$ of a QTA. The usual formulation of effective bandwidth implicitly pays the urgency debt by under-utilisation of the interface in question, by transforming it into a required dedicated capacity for the stream.

### 4.1.2. Cost of Servicing Bursty Traffic

The above example, of low data rates with strict (above the natural urgency) requirements on the $\lceil \Delta Q \rceil$, might typify classes of IoT. Were such IoT use-cases to become a significant fraction of the total load, their collective urgency requirements would create a significant economic cost related to their timely end-to-end delivery.

IoT is not the only situation where such cost of burstiness arises. Situations arise in highly interactive virtualised gaming environments where the UX is predicated on a low, and consistent, $\lceil \Delta Q \rceil$; requirements for click-to-eyeball response times < 50 ms are not unusual. After accounting for game server processing costs and $\Delta Q_{|G,S}$ (structural delay), just 3–5 ms remain for the forwarding of a 128 kB video frame at key contention points. This urgency requirement demands an allocation of 5 to 10 times the average bandwidth (which is $\approx$40 Mbps). This has commercial implications, as that capacity also has to be present in the last mile connection for such deployments.

### 4.1.3. Urgency, Utilisation and Schedulability

Even when capacity is not overbooked, it is possible that, when sufficient flows require a $\Delta Q$ that is better than their natural urgency, the collective demand for timely processing can no longer be met. This situation could occur due to inherent randomness in arrival patterns and/or effects of correlated demand. (By contrast, SDH/PDH networks could be viewed as communication infrastructure that only supports data transport outcomes at their natural urgency, with enforced out-of-phase correlation.) When this is acute, schedulability may require deliberate under-utilisation, as even the residual service time effects of processing those flows that have no requirement on their urgency can consume too much of the $\Delta Q$ budget.

RINA (with its richer, quality-aware routing) can exploit disjoint paths, so traffic that has lower urgency requirements can be routed via paths with larger structural $\Delta Q$ [15].

There is an analogy here with the notion of "Orders of Hazard" (Table 2), namely:

**0th order: Causality** The required $\lceil \Delta Q \rceil$ is not attainable over the end-to-end path.
**1st order: Capacity** The loss bounds of $\Delta Q$ cannot be delivered as the capacity along the end-to-end path is insufficient.

**2nd order: Schedulabilty** The combined urgency requirements under normal operation cannot be met.

**3rd order: Internal Behavioural Correlations** There are correlations (under the control of the system operator) that cause aggregated traffic patterns such that their collective $\lceil \Delta Q \rceil$ can no longer be met.

**4th order: External Behavioural Correlations** As above, but where the external environment imposes the correlations.

The wider implication of having this fine-grained understanding of the interaction between network resource configuration and delivered outcomes is that it allows for higher level systemic optimisation. This, in turn, helps manage both the quantity and timescale of capital expenditure.

### 4.1.4. Flow Admission

Managing the hazard of breaching a QTA due to overbooking requires the ability to reject new loads and even shed existing ones. This is typically done at a high layer in the protocol stack, such as an HTTP server rejecting connections, but such approaches still allow lower-level resources to be consumed. RINA, on the other hand, provides policy hooks to perform this at *every* layer based on an assessment of the QTA breach hazard.

### 4.1.5. Traffic Shaping

On very short timescales, even conservatively allocated resources can be overwhelmed by bursts of load that expose the limits of schedulability. These can be ameliorated by including burst limits in the QTAs, and shaping arriving load to fit [15]. This constrains the schedulability risk to that induced by overbooking.

### 4.2. Timescales of Management

Many current distributed systems attempt to manage performance using end-to-end control loops [18], whose time-constants are necessarily orders of magnitude larger than those of phenomena such as burst-induced scheduling problems. Such loops are therefore unable to manage short-term quality attenuation; at best, they can constrain overbooking of capacity, provided the applied load is stable over sufficient multiples of their time-constant.

RINA provides control of scope and therefore bounds the time-constants of cross-DIF control loops. No control loop can operate quickly enough to deal with instantaneous bursts, however, and so it is essential to also have a resource scheduling mechanism that is unconditionally stable [19], even under sudden drops in available resources [20]. This gives enough time for flow control mechanisms to manage elastic demand sources and flow-based admission/rejection mechanisms to respond to maturing of overbooking risks if the situation persists. For example, this can be done by signalling the change in attenuation via the RIB, and/or altering the congestion management policy. In extremis, low-precedence flows can be forced to re-engage in the allocation process; the higher layer DIF can then apply its own policy, such as finding an alternative lower-layer DIF that can satisfy its QTA.

### 4.3. Supply Variation

In order to detect variation in supply, it is necessary to measure the quality attenuation delivered as well as the transported load. If quality attenuation is seen to increase without a corresponding increase in the transported load (i.e., this is not due to increase in demand), there are a number of potential responses that a policy could adopt:

1. Adapt the local scheduling policy so as to concentrate the performance hazard into less important traffic (not necessarily the traffic with the largest attenuation tolerance.), for example favouring traffic required for maintaining network stability;

2. In the case of loss variability, one option is to renegotiate the loss concealment strategy of connections achieving lower loss at the cost of increased load and/or delay (e.g., using Fountain codes [21]).

3. Seek additional resources in the form of new/alternative lower-level DIFs, similar to the classic solution to mobility [4]; the difference here is that a lower-layer DIF may be offering connectivity and even capacity, but be unable to satisfy the quality attenuation bounds, forcing an alternate route to be sought.

### 4.4. Demand Variability

One of the central challenges in managing performance is anticipating shifts in demand. The associated-by-default assumption of IP networks makes this very difficult, at least at the IP level, as any source can legitimately send traffic to any destination at any time (this is of course a feature that enables DDoS attacks). RINA offers considerable benefits here, as the limited scope of a DIF and the requirement to enrol in a DIF provides a bound on the population of potential sources. Conversely, the potential for very long-lived associations within upper layer DIFs with no currently active connections represents a hazard that must be tracked. Equally, the number of currently active connections gives visibility of the immediate level of load.

Depending on the 'openness' of the system, one issue is that of incentivising accurate characterisation of demand, to minimise the risk of arbitrage (or at least wasted resources). This requires some means of associating costs with QTAs, which can then be managed with approaches such as tangent pricing [22]. Analysis of the economic/game-theoretic mechanisms for long-term system stability is out of scope here.

### 4.5. Correlated Load

We can define a hierarchy of 'orders of performance hazard', corresponding to increasing sophistication of performance hazard management, as set out in Table 2.

There are two sources of correlated load:

1. Internal correlations due to the behaviour of the system (for example, sending a large SDU creates a burst of lower-layer PDUs);

2. External correlations due to the behaviour of the users of the service (for example, an early evening 'busy hour').

Internal correlations should be resolved by analysis of the system design and testing during implementation. Externally correlated load, whether due to normal usage or malicious behaviour, is the most difficult hazard to manage, especially for an autonomous system. Note, this may be a legitimate application for machine learning techniques, using the rich information available in the RIB.

**Table 2.** Orders of Hazard.

| Order | Subject of Concern |
|---|---|
| 0: Causality | Does best case low-level $\Delta$Q permit the system to deliver any successful top-level outcomes within the $\Delta$Q demanded? |
| 1: Capacity | Will the delivered $\Delta$Q be within the requirement at economic/expected levels of load? |
| 2: Schedulability | Can the QTAs be maintained during (reasonable) periods of operational stress? |
| 3: Behaviour | Does the system contain (or is it sensitive to) internal correlation effects/events: how does that influence QTA breach hazards? |
| 4: Stress | Is the system sensitive to load correlation effects created outside the system(s) under consideration? |

## 5. RINA and Performance Management

As touched on in previous sections, RINA provides ways to mitigate performance hazards, with capabilities to manage the full resource life-cycle over a wide dynamic range. The acquiring, allocating and releasing of resources can be achieved using in-band mechanisms that provide immediate feedback. This contrasts with attempts to do this within IP, e.g., DiffServ, which defines a packet marking that drives an implicit resource allocation mechanism, but has no means of knowing the emergent results. RINA's recursive architecture automatically invokes a nested set of policy responses that enables distributed orchestration of resources to optimise the global system configuration.

The RINA DIF model defines constrained universes of discourse, making performance management tractable.

### 5.1. Connection Life-Cycle and QTAs

Enrolling in a DIF provides not only a set of fixed concrete representations and protocol choices (e.g., authentication models), it also supplies a set of policy operations. The 'richness' (and indeed the complexity) of the DIF can be tailored to the collective requirements. This contrasts with today's Internet where the implicit aim is to create unconstrained universality, delivering a lowest-common-denominator form of networking.

Within the operation of a DIF, the relationships between end points have clear and specified phases.

### 5.1.1. Allocation and Call Admission Control

The allocation phase of a RINA connection provides the opportunity to perform appropriate Call Admission Control (CAC) [23]. The allocation process discovers the location of a destination, and the associated reply establishes a binding from the name to an address. Each allocation presents a QoS Cube (which can be viewed as a concrete representation of a QTA) to the DIF. At the end of this discovery process, there is now sufficient information to make routing decisions. In particular, it provides both the potential load at an interface and the collective constraints on the $\Delta Q$ for the potential flows through it: the set of the QTAs at this interface. The information is available to configure queuing, policing, and scheduling mechanisms, and to inform overbooking policies and estimate the associated performance hazards.

### 5.1.2. Allocation and Slack/Hazard

RINA uses the $\delta t$ data protocol model [24]. With this approach, it is possible for the state associated with connection to be discarded when there has been no data transfer activity for a while; provided an allocation exists, it is sufficient to use the next packet to re-establish the required state. The existence/non-existence of such state gives an indication of the periods of activity on a connection. This also gives information to quantify the probability (and severity) of performance hazards associated with overbooking. This is discussed more in Section 6.1 below.

### 5.1.3. Active Data Transport and Scheduling

Active Data Transport is the period during which policing, shaping, queuing and scheduling of the constituent packet flow actually occurs - these being the mechanisms by which $\lceil \Delta Q \rceil$ is ensured, and over which the short-term trading of $\Delta Q$ actually occurs.

RINA also provides a Resource Information Base (RIB), with a uniform interface against which both static and dynamic information can be recorded and disseminated. Such a capability would be universal and ubiquitous in a RINA deployment. This can give a system operator the real-time views with which they could go beyond simply assuring adherence to performance constraints (by measuring $\Delta Q$ and comparing against QTAs) towards optimisation of performance/cost based on the more global view that the RIB supports.

## 6. Performance Management Metrics

There are some key indicators that can be measured, both as inputs to a autonomous control algorithm and as external metrics of the effectiveness of the performance management system.

### 6.1. ΔQ Slack/Hazard

Comparing the delivered ΔQ with that required by the QTA may be reducible to a simple parameter in some cases. Figure 6 shows an example of a simple ΔQ budget (shown in blue): 90% of outcomes should occur within 50 ms. The upper black line shows a delivered ΔQ, which is strictly smaller (by the definition of 2). The green lines show one way the difference could be converted into a number, by taking the area between them and the delivered ΔQ. This represents a measure of 'slack' in the delivery of the requirement.

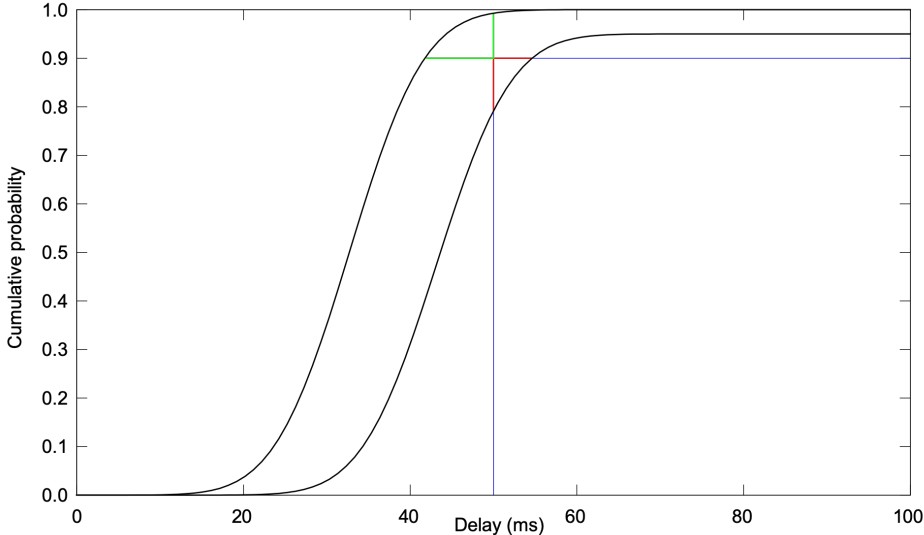

**Figure 6.** QTA breach slack/hazard.

The lower black line shows delivered ΔQ that is not smaller than the requirement. In this case, the area enclosed by the red lines and the ΔQ curve represents a measure of the performance 'hazard'. Extending these simple cases into a general measure is a subject for further research, but having such a measure would quantify how 'tightly' QTAs are being met or how severely they are being broken.

Figure 7 shows measurements of just such a QTA breach metric on a live network over a period of several days (thanks to the Kent Public Service Network for permission to use this). The red crosses show the breach metric (in arbitrary units), as derived from measured ΔQ (using one of the procedures defined in Section 3.1), and the green line shows link utilisation over the same timescale, by time of day over a sample week. It is tempting to see this as evidence of a particular causality—that high link utilisation 'causes' QTA breach. However, there are definite counterexamples to this within this data: there is a point (early hours of the 1st June) in which the breach is present while the overall utilisation is low (actually <0.1%), whereas the highest utilisation peaks (early evening of the 2nd, 3rd, 4th, and 5th) do not have any associated QTA breaches. There is a correlation, but its causality is different: frequent QTA breaches represent a high *instantaneous* load, which contributes to raising the *average* utilisation.

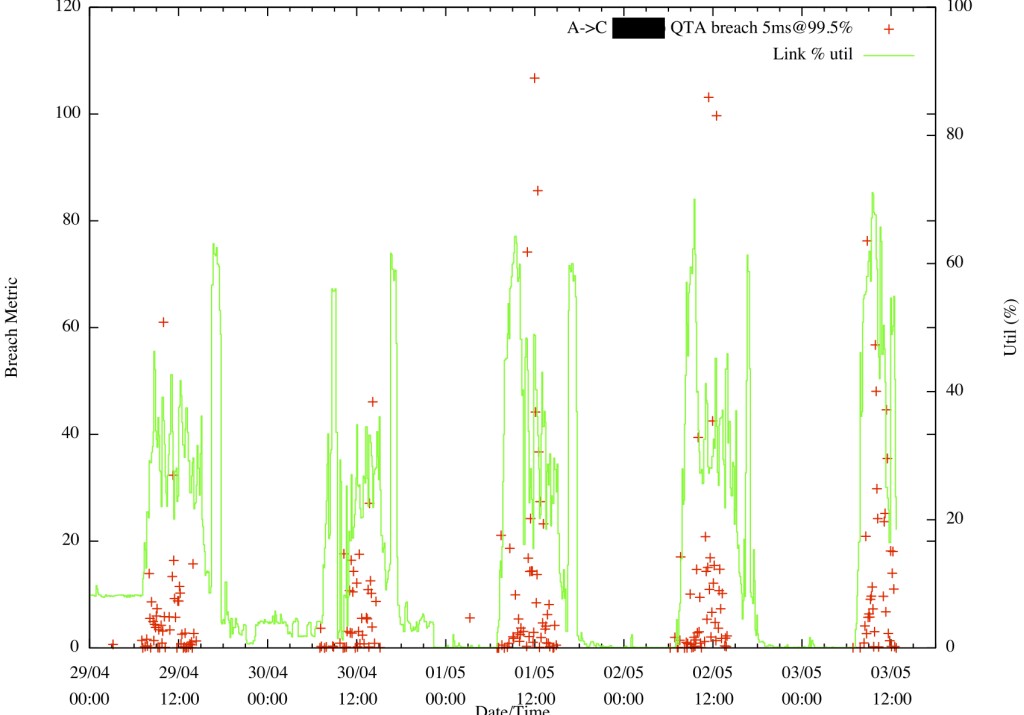

**Figure 7.** Correlation of QTA breach with load.

## 6.2. Capacity Utilisation

A further measure of the effectiveness of performance management is the extent to which low-level capacity is being utilised. Effective scheduling makes it possible to satisfy QTAs while maintaining high utilisation [19], but managing performance hazards may require spare capacity to cope with variation in load. If the capacity used can be varied, for example in a dynamic network slicing context, then the effectiveness of the performance management can be assessed by how tightly this resource can be managed.

## 6.3. Coefficient of Variation

One oft-cited problem in large networks is the tendency for traffic to become highly bursty, potentially having aspects of self-similarity [25]. This is clearly problematic for delivering bounds on $\Delta Q$, and can be thought of as a type of self-induced load correlation. The coefficient of variation of the arrival pattern is one measure of the burstiness of the load. Given appropriate (non-work-conserving) shaping capabilities, the performance management system can trade increased $\Delta Q$ at one point (by introducing shaping delays) for a lower coefficient of variation at a downstream multiplexing point to move towards a better global configuration for the system as a whole.

A good outcome for performance management would be to ensure that the coefficient of variation does not increase as traffic flows are aggregated, so as to permit efficient scheduling at all points along the path.

## 7. An Outline Architecture

What has been presented here is a system for quantifying performance based on quality attenuation and QTAs. Any system has a finite capability, both over long timescales (the standard issue of capacity) and over short ones (the question of schedulability that we have highlighted in this paper). A performance management system is the means by which these limited resources are allocated (including deciding the winners/losers in times of excess contention). How the allocation is decided is a separate question, which is a subject for further research.

QTAs come from the QoE requirements for a given application to support a given role: the same application, used in a different role, might require a different QTA/SLA (e.g., compare the requirements for VoIP between its typical and safety-of-life uses). Note that the ΔQ requirements to support an application may also depend on additional constraints: consider a VoIP call between local endpoints vs. one that is intercontinental; or an application in which much of the end-to-end time budget is consumed by computation or I/O (as discussed in Section 4.1.2). In a large-scale mixed-use environment, end-users will subscribe to an application DIF/DAF, which will negotiate with the network infrastructure (possibly including multiple providers) to obtain QTA/SLAs to satisfy the overall end-user requirement. Given that the end-to-end path has been allocated (see Section 5.1.1), there will be an associated ΔQ, in particular, there will be structural ΔQ ($\Delta Q_{|G,S}$) and an overall path-related variability ($\Delta Q_{|V}$). This is precisely the scenario as discussed in [26], and the potential for real-time trading and optimisation now arises.

Managing performance of a large-scale distributed system requires a coordinated application of a number of mechanisms that operate over different timescales:

**On short timescales,** this is a matter of distributing the quality attenuation due to resource sharing according to the requirements of a set of QTAs. This is the domain of queuing and scheduling mechanisms, which must deliver predictable outcomes regardless of the instantaneous load [19].

**On medium timescales,** it involves:

1. Selecting appropriate routes for traffic and configuring queuing and scheduling mechanisms according to the current set of requirements [15];
2. Managing demand so that such configuration is feasible. For elastic flows, this additionally means managing congestion with protocol control loops, and for inelastic flows controlling admission and, if necessary, gracefully shedding load while maintaining the critical functions of the system;
3. Measuring performance hazards, particularly those resulting from overbooking and load correlation, and adjusting routing and scheduling configurations to minimise them. Where performance hazards result in QTA breaches, this should be recorded and related to service-level agreements that specify how frequent such breaches are allowed to be.

RINA supplies both the required control mechanisms and signalling (e.g., via QoS-cubes) so that higher network layers and, ultimately, applications, can adapt.

**Over longer timescales,** there are options for RINA management functions to trade quality attenuation budget between elements whose schedulability constraints vary, assuming some stationarity of applied load and a scheduling mechanism within the elements that allows trading of ΔQ between flows at a multiplexing point. This would allow an overbooking hazard to be spread more evenly, and avoid rejection of new flow requests just because a single element along the path is constrained [27].

**On the longest timescales,** it requires provisioning new capacity; requests for this could be generated automatically, although fulfilling them might involve manual processes.

RINA naturally embeds many of these capabilities into the layer management functionality as discussed in Section 5. The extent to which operational choices are made locally or passed to a management function with a wider scope becomes a matter of management policy, permitting a spectrum of more centralised vs. more distributed implementations.

## 8. Conclusions

### 8.1. Requirements for Delivering Performance

We have refined the vague notion of 'delivering good performance' for a service into the precise requirement of maintaining bounded quality attenuation. Achieving this is a matter of matching

supply (in the form of various resources) and demand (applied load and required quality attenuation) over various timescales, which maps well into the layered scope provided by RINA. We have also proposed a set of metrics for measuring the effectiveness of performance management.

### 8.2. Managing Performance at Scale

Delivering effective scalable performance management requires the ability to aggregate and dis-aggregate requirements and emergent performance both 'vertically' (between different layers) and 'horizontally' (between different functional elements). RINA provides the mechanism for defining the appropriate scopes and conveying the information, and $\Delta Q$ provides the calculus for performing the operations.

### 8.3. Benefits of Performance Management

Deployment of a RINA-based solution may appear problematic, given the absence of large-scale deployment of RINA itself (although there are movements in this direction e.g., [28]). However, RINA is highly suited to an 'underlay' deployment, transparent to IP-based applications [3]. Application areas where the insecurity and unpredictability of current all-IP solutions are problematic, and where a single authority could take the decision to exploit RINA, are the most likely entry points for such a solution. A network operator, whether a service provider or enterprise, etc., could use this approach to deliver high-value high-integrity IP-based services.

Increasing autonomy of large-scale distributed applications reduces the cost of deploying and maintaining them, and performance management increases their reliability. For systems delivering public services, this should, in time, reduce their cost and improve the level of trust in them. For those delivering private or commercial services, it should increase the value of the service and hence the revenue potential.

It is also worth noting that there is an interaction between performance management and some aspects of security. The fact that $\Delta Q$ is conserved means that it provides a check on certain system invariants. For example, severe traffic mis-routing [29] would immediately raise performance alarms, since there is no way to disguise the fact that $\Delta Q_{|G}$ suddenly increases.

### 8.4. Directions for Future Work

There is clearly a substantial effort required to turn the ideas in this paper into a practical system. A first step would be to articulate the components of a RINA performance management system and the information exchange requirements between them. This would pave the way for specifying layer management policies for performance management of future RINA systems.

Means of establishing the QTAs for applications need to be developed, so as to provide appropriate input to the performance management system. Another important area is to define more robust and practical measures of performance hazards on which the system can act.

To move from internal/private use of such networks to external/public use will require mechanisms to impose either economic (dis)incentives or rationing to prevent abuse. Either approach requires measurement of usage, both in terms of what applications request from the network and what they actually consume. RINA's layered structure and bounded scopes offer a framework for gathering and using such information in an efficient and scalable fashion.

Finally, the concepts discussed here provide a lexicon for discussing the shortcomings of the current situation. Questions of characterising both demand and supply and of communicating requirements are universal.

**Author Contributions:** Conceptualization, N.D. and P.T.; formal analysis, N.D.; writing—original draft preparation, N.D. and P.T.; writing—review and editing, N.D. and P.T. All authors have read and agreed to the published version of the manuscript.

**Funding:** This research received no external funding.

**Acknowledgments:** The authors would like to thank Lucy Hazell for assistance with reviewing the manuscript.

**Conflicts of Interest:** The authors declare no conflict of interest.

## Abbreviations

The following abbreviations are used in this manuscript:

| | |
|---|---|
| ΔQ | Quality Attenuation |
| DDoS | Distributed Denial of Service |
| DIF | Distributed Interprocess communication Facility |
| PDU | Protocol Data Unit |
| IRV | Improper Random Variable |
| QTA | Quality Transport Agreement |
| RIB | Resource Information Base |
| RINA | Recursive InterNetworking Architecture |
| SDN | Software Defined Networking |
| SDU | Service Data Unit |
| SLA | Service Level Agreement |
| UX | User eXperience |

## Appendix A. Worked Example of Composition of Quality Attenuation

As an example, we consider a very generic Remote Procedure Call (RPC) in which a 'front-end' (e.g., web browser, embedded sensor, smart meter) performs a transaction across a network with a 'back-end', where the transaction includes an interaction with a database of some sort. This is illustrated in Figure A1, which shows the system components in blue and the typical sequence of events as numbered circles. We measure the performance of the system on the basis of passage times between the labelled observation points A–F, which we characterise using improper random variables, ΔQ.

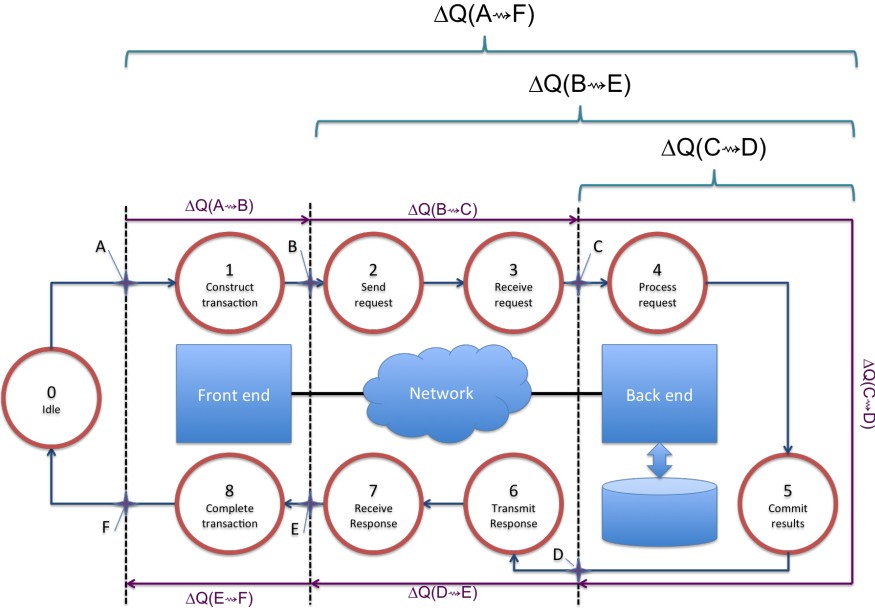

**Figure A1.** Steps of an RPC with observation points.

From a quality attenuation perspective, how the transition from C to D occurs is irrelevant; all that matters is how long it might take and how likely it is to fail (its ΔQ). Thus, from the viewpoint of the rest of the system, we can 'roll up' the last stage of the process into a ΔQ. In the same way, we can combine the performance of the network portions with that of the back-end to give a composite ΔQ between observables B and E (this 'rolling up' process is shown at the top of the figure). Finally, we

can incorporate the front-end behaviour to give the ΔQ for the whole transition between observables A and F. This is the quality impairment from the 'user' point of view, which is where we begin in imposing requirements.

For this example, we express the performance requirement between observables A and F as the proportion of responses that should occur within a specific time bound:

- 50% within 500 ms;
- 90% within 700 ms;
- 95% within 1 s;
- 99% within 2 s;
- 1% chance of failure to respond within 2.5 s, considered a failed RPC.

How the 'rolled up' ΔQ of the remainder of the system impacts the performance of the front-end component depends on its behaviour, which we model as a simple retry protocol. Since the request or its acknowledgement may be lost by the network, the process retries after a timeout (which we set to be 0.33 s), and after N attempts the transaction is deemed to have failed. This can be represented with a simple state diagram (with transitions annotated with the corresponding observables) in Figure A2.

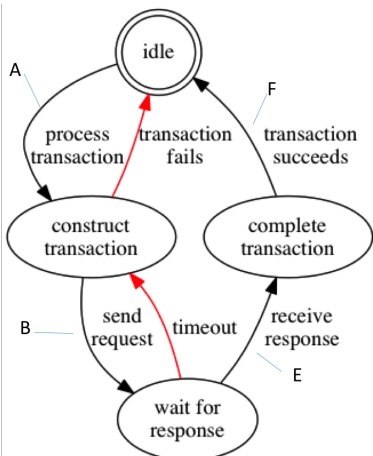

**Figure A2.** State transition diagram of RPC front-end.

If N = 4, we can unroll the process as in Figure A3. There is a non-preemptive first-to-finish synchronisation between receiving the response and the timeout; which route is taken is a probabilistic choice, shown here with blue and green examples. The path taken depends on the ΔQ of the B–E path, encapsulated here in the 'wait for response' states. We combine the ΔQs of the different paths with these probabilities.

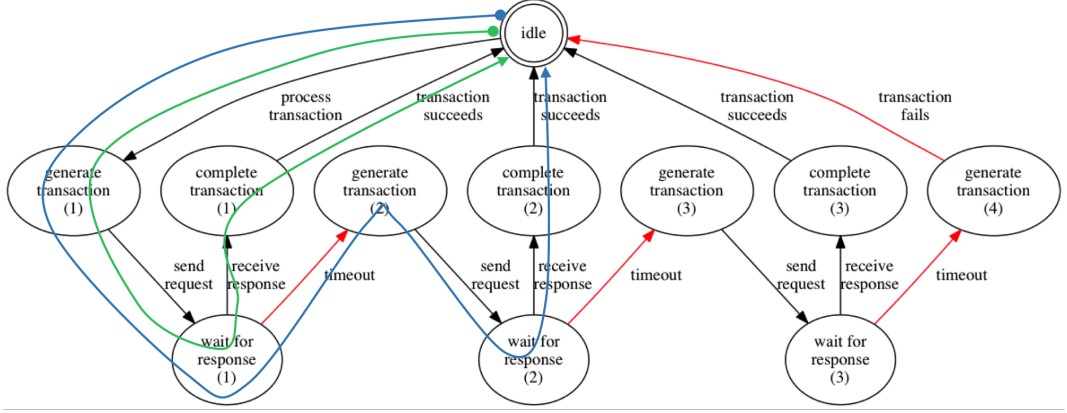

**Figure A3.** Unrolled state transition diagram.

For this example, we consider a specific scenario in which the front end is in the UK, connected by typical broadband, and the server is in a datacentre on the US East Coast. Thus, for the network transmissions, we set $\Delta Q(B \to C)$ and $\Delta Q(D \to E)$ to be the same, with a minimum delay of 95 ms, 50 ms variability and 1–2% loss. For the server, we assume $\Delta Q(C \to D)$ is uniformly distributed between 15 ms and 30 ms. Plotting these distributions as CDFs gives Figure A4a.

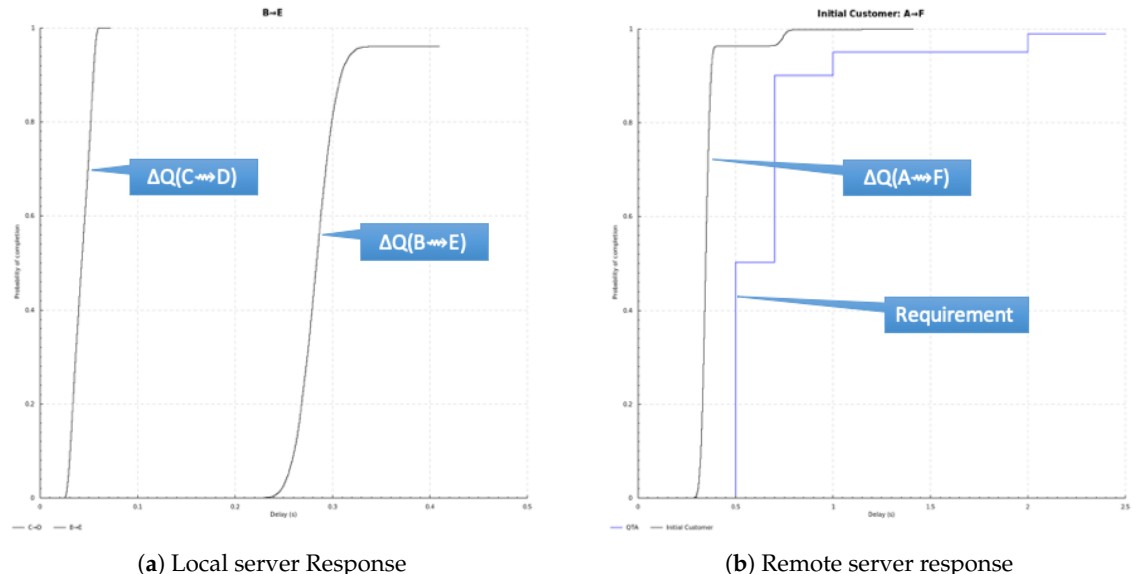

(**a**) Local server Response        (**b**) Remote server response

**Figure A4.** Performance CDFs of the first scenario.

The leftmost curve of Figure A4a shows the server response, and the rightmost curve of of the same figure shows the same distribution convolved with the network transmission $\Delta Q$ (twice, once for each direction). This gives the overall $\Delta Q$ for the remainder of the system, which is sampled by the front-end process in its 'wait for response' state. Combining this with the probabilistic choice of paths through the front-end state space gives the first CDF in Figure A4b, which is the full distribution of the delay (and the probability of failure) between observables A and F. This is plotted against the system requirement for comparison. It can be seen that the result curve is everywhere to the left and above of the requirement, which is the condition for the requirement to be met, as previously discussed in Section 2. This confirms that the performance requirements are met for the network and server subsystems.

Thus far, the verification task seems straightforward: we have seen that, provided the network and server performance are within the given bounds, then the front end will deliver the required end-user experience. However, this seems to have plenty of slack, which naturally leads to the question of how far we can relax the requirements on the performance of the subsystems.

We first consider a different scenario in which the server performance is the same, but the network has varying delays, due to having an alternate server on the US West Coast. This is modelled as a probabilistic choice between different delay distributions, with a 10% chance of choosing the more distant server, with twice the network delay of the nearer one. This gives the distributions shown (as CDFs) in Figure A5a.

The leftmost curve of Figure A5a shows the server response as before, and the middle curve the combined network/server delay of the original scenario. The rightmost curve of A5a is the result of convolving the server response with the revised network $\Delta Q$.

We can see in Figure A5b that the overall $\Delta Q$ resulting from combining this with the choices in the front-end state machine (Figure A3) no longer meets the requirement.

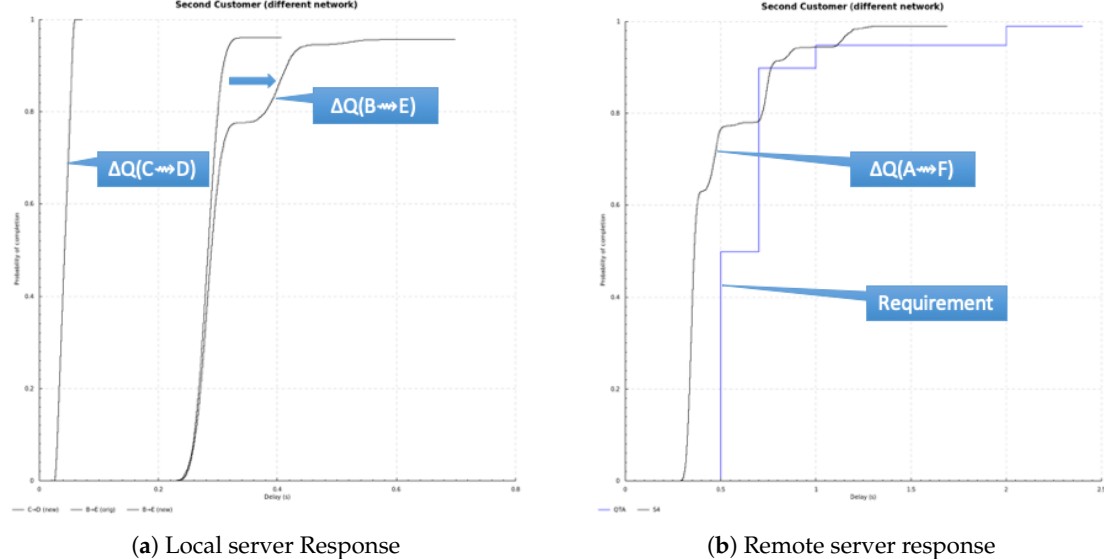

(**a**) Local server Response　　　　　　　　　　　　　　(**b**) Remote server response

**Figure A5.** Response CDF for physically distributed alternate servers.

Secondly, we can consider another scenario in which the network performance is unaltered but the server performance varies. If the server is virtualized, there will typically be a limit on the rate of I/O operations it can perform, so that the underlying infrastructure can be reasonably shared. Typically, access is controlled via a token-bucket shaper that allows small bursts while limiting long-term average use. The delivered performance of the virtualized I/O thus depends on recent usage history. If we now consider that there will be a number of front-ends accessing the server, the overall load will be a function of the number of active front-ends and the proportion of retries sent. Re-tries depend on the $\Delta Q(B \to E)$, which in turn depends on the performance of the server, and hence of the I/O system. Modelling this fully would require iterating to a fixed point for the whole system, and is beyond the scope of this paper, so for simplicity it is represented here by a probability of taking a 'slow path' in the server, which has a uniform distribution of response times between 15 ms and 150 ms. In Figure A6a, the leftmost curves are the CDFs of the server response as the probability of taking the slow path varies from 0% to 25%, 50%, 75%, and 100%, and those on the right are the corresponding set of CDFs of the server response convolved with the original network performance.

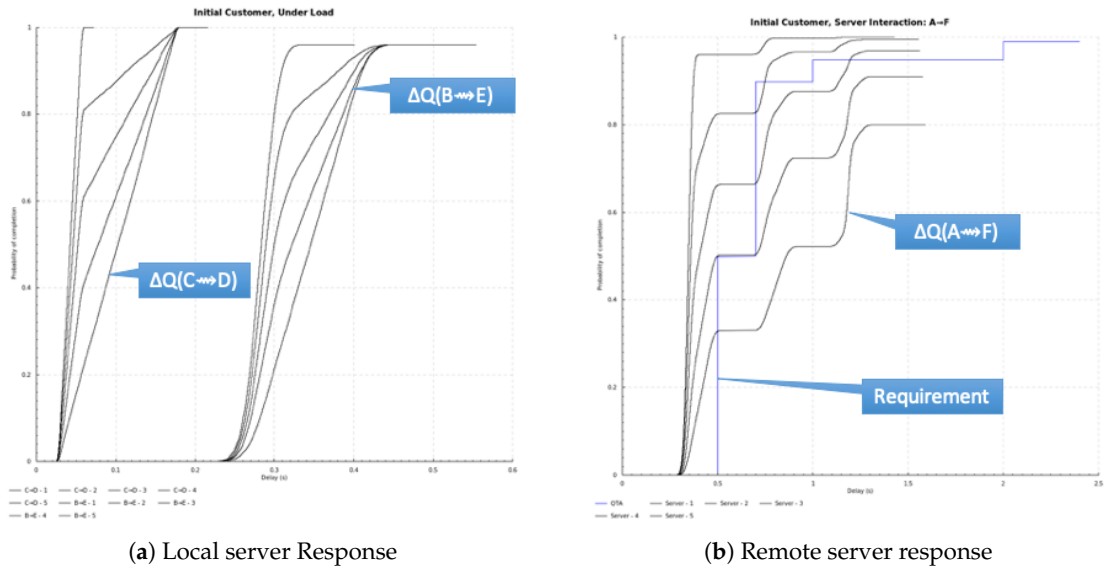

(**a**) Local server Response　　　　　　　　　　　　　　(**b**) Remote server response

**Figure A6.** Response CDF with potential use of server slow-path.

Figure A6b shows the result of passing these distributions through the with the choices in the front-end state machine (Figure A3), with the original requirement shown for comparison. It can be seen that none (excluding that of 0% chance of taking a slow server I/O path) of the curves fully meets the requirement.

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
