# Peer review of "Towards a RINA-Based Architecture for Performance Management of Large-Scale Distributed Systems†"

_computers, doi:10.3390/computers9020053_

Round 1

Reviewer 1 Report

The authors present a great work about Recursive Internetworking Architectura, they present metrics to test the performance management system. The paper is very well structured and sounds good. The example included in the appendix is very illustrative of the proposal.

Just a few comments to improve the article in a simple way.

Q1. Lines 68-69, 118-128. Here you can use a latex itemize type structure (same as those on lines 149-151).
Q2. Line 105. Any reference to improper random variables (IRV) is welcome.
Q3. Conclusions. Some reference to SDN could be included in the future lines in the sense included in:
https://sdn.ieee.org/newsletter/january-2017/progressive-network-transformation-with-rina

Q4. Conclusions. Is there any aspect of metric security as future work?
Q5. References. There are few references and some are old. Some recent examples that could be included are:

Tarzan, M., Bergesio, L., & Grasa, E. (2019, February). Error and Flow Control Protocol (EFCP) Design and Implementation: A Data Transfer Protocol for the Recursive InterNetwork Architecture. In 2019 22nd Conference on Innovation in Clouds, Internet and Networks and Workshops (ICIN) (pp. 66-71). IEEE.

Welzl, M., Teymoori, P., Gjessing, S., & Islam, S. (2020, February). Follow the Model: How Recursive Networking Can Solve the Internet’s Congestion Control Problems. In 2020 International Conference on Computing, Networking and Communications (ICNC) (pp. 518-524). IEEE.

Thompson, P., & Davies, N. (2020, February). Towards a performance management architecture for large-scale distributed systems using RINA. In 2020 23rd Conference on Innovation in Clouds, Internet and Networks and Workshops (ICIN) (pp. 29-34). IEEE.

Author Response

Q1. Lines 68-69, 118-128. Here you can use a latex itemize type structure (same as those on lines 149-151).

This has been changed.

Q2. Line 105. Any reference to improper random variables (IRV) is welcome.

A reference has been added.

Q3. Conclusions. Some reference to SDN could be included in the future lines in the sense included in:
https://sdn.ieee.org/newsletter/january-2017/progressive-network-transformation-with-rina

Q4. Conclusions. Is there any aspect of metric security as future work?

Additional text has been added to the conclusion. See attached version.

Q5. References. There are few references and some are old. Some recent examples that could be included are:

Additional references have been included.

Reviewer 2 Report

The article is interesting and educational. It is long. However, it is difficult for many readers who are unfamiliar with RINA and J Day's arguments to understand.
To increase the clarity of the article, it is essential to add a paragraph on RINA. This short paragraph is missing to remind what RINA is and why the QoS of an IP network is fuzzy and impossible to manage correctly and why it is possible to define new and more precise concepts of QoS with RINA.
Everything would be clear in this article, if we add the originality of RINA: RINA is not like TCP by joining ports on a network, but RINA connects distributed processes. So everything is explained since the network no longer carries bits but relates processes: it is therefore possible to design QoS which regulates all kinds of applications, including those that request local or global network timing.
It also lacks a paragraph of conclusions at the end of the article to explain the future of this type of approach: fans of TCP / IP and the old internet can not imagine that we dare to drastically improve the internet. It is also important to explain how we can deploy this type of network: in a company, with an operator, etc.

Author Response

The article is interesting and educational. It is long. However, it is difficult for many readers who are unfamiliar with RINA and J Day's arguments to understand.
To increase the clarity of the article, it is essential to add a paragraph on RINA. This short paragraph is missing to remind what RINA is and why the QoS of an IP network is fuzzy and impossible to manage correctly and why it is possible to define new and more precise concepts of QoS with RINA.
Everything would be clear in this article, if we add the originality of RINA: RINA is not like TCP by joining ports on a network, but RINA connects distributed processes. So everything is explained since the network no longer carries bits but relates processes: it is therefore possible to design QoS which regulates all kinds of applications, including those that request local or global network timing.

A paragraph discussing RINA has been added to the introduction.

It also lacks a paragraph of conclusions at the end of the article to explain the future of this type of approach: fans of TCP / IP and the old internet can not imagine that we dare to drastically improve the internet. It is also important to explain how we can deploy this type of network: in a company, with an operator, etc.

Additional material has been added to the conclusion - see attached version.
